# Phase I Study of a Multivalent WT1 Peptide Vaccine (Galinpepimut-S) in Combination with Nivolumab in Patients with WT1-Expressing Ovarian Cancer in Second or Third Remission

**DOI:** 10.3390/cancers15051458

**Published:** 2023-02-25

**Authors:** Beryl L. Manning-Geist, Sacha Gnjatic, Carol Aghajanian, Jason Konner, Sarah H. Kim, Debra Sarasohn, Krysten Soldan, William P. Tew, Nicholas J. Sarlis, Dmitriy Zamarin, Sara Kravetz, Ilaria Laface, Teresa Rasalan-Ho, Jingjing Qi, Phillip Wong, Paul J. Sabbatini, Roisin E. O’Cearbhaill

**Affiliations:** 1Gynecology Service, Department of Surgery, Memorial Sloan Kettering Cancer Center, New York, NY 10065, USA; 2Immune Monitoring Facility, Memorial Sloan Kettering Cancer Center, New York, NY 10065, USA; 3Tisch Cancer Institute, Precision Immunology Institute, Human Immune Monitoring Center, Icahn School of Medicine at Mount Sinai, New York, NY 10029, USA; 4Department of Medicine, Weill Cornell Medical Center, New York, NY 10065, USA; 5Gynecologic Medical Oncology Service, Department of Medicine, Memorial Sloan Kettering Cancer Center, New York, NY 10065, USA; 6Department of Radiology, Memorial Sloan Kettering Cancer Center, New York, NY 10065, USA; 7SELLAS Life Sciences Group, Inc., New York, NY 10036, USA; 8Department of Medicine, University of Galway, H91 YR71 Galway, Ireland

**Keywords:** immunotherapy, antigens, vaccine, immunogenicity, programmed cell death 1 receptor, epithelial ovarian cancer

## Abstract

**Simple Summary:**

Approximately 70% of patients with advanced epithelial ovarian cancer who achieve clinical remission after initial surgery and chemotherapy have a recurrence. Wilms’ Tumor 1 (WT1), which is overexpressed in ovarian cancer cells, is a promising target for tumor-directed immunotherapy for ovarian cancer due to its prevalence and specificity. The aim of our open-label, non-randomized phase I study was to assess the safety of a WT1 peptide vaccine (galinpepimut-S) in combination with nivolumab in patients with WT1-expressing ovarian cancer in second or third remission. In a sample of 11 patients, the combination of galinpepimut-S vaccine and nivolumab induced immune responses and was deemed safe and tolerable. Our findings provide additional evidence that the combination of immune checkpoint inhibitors (e.g., nivolumab) and vaccines results in enhanced anti-tumor immune responses.

**Abstract:**

We examined the safety and immunogenicity of sequential administration of a tetravalent, non-HLA (human leukocyte antigen) restricted, heteroclitic Wilms’ Tumor 1 (WT1) peptide vaccine (galinpepimut-S) with anti–PD-1 (programmed cell death protein 1) nivolumab. This open-label, non-randomized phase I study enrolled patients with WT1-expressing ovarian cancer in second or third remission from June 2016 to July 2017. Therapy included six (every two weeks) subcutaneous inoculations of galinpepimut-S vaccine adjuvanted with Montanide, low-dose subcutaneous sargramostim at the injection site, with intravenous nivolumab over 12 weeks, and up to six additional doses until disease progression or toxicity. One-year progression-free survival (PFS) was correlated to T-cell responses and WT1-specific immunoglobulin (Ig)G levels. Eleven patients were enrolled; seven experienced a grade 1 adverse event, and one experienced a grade ≥3 adverse event considered a dose-limiting toxicity. Ten (91%) of eleven patients had T-cell responses to WT1 peptides. Seven (88%) of eight evaluable patients had IgG against WT1 antigen and full-length protein. In evaluable patients who received >2 treatments of galinpepimut-S and nivolumab, the 1-year PFS rate was 70%. Coadministration of galinpepimut-S and nivolumab demonstrated a tolerable toxicity profile and induced immune responses, as indicated by immunophenotyping and WT1-specific IgG production. Exploratory analysis for efficacy yielded a promising 1-year PFS rate.

## 1. Introduction

Approximately 70% of patients with advanced epithelial ovarian cancer (EOC) who achieve clinical remission after initial surgery and chemotherapy will recur. Over a disease course, which typically entails successive recurrences, patients may require multiple lines of cytotoxic, hormonal, and/or targeted therapy [1]. With each treatment, remission duration usually shortens until broad chemoresistance develops [1], highlighting the need for novel strategies to extend remission duration and optimize survival outcomes.

There are ongoing trials to determine the safety and efficacy of immune-oncology therapies, such as immune checkpoint inhibition, adoptive cellular therapies, and cancer vaccines, in the treatment of recurrent EOC. Preclinical models have demonstrated efficacy with both passively administered antibodies and vaccines, suggesting a role for immunomodulation in EOC treatment [2,3]. Furthermore, higher levels of tumor-infiltrating T cells are associated with longer survival in patients with EOC [4,5,6]. Despite this compelling evidence, investigations have failed to demonstrate the benefit of immunomodulatory agents in EOC treatment, and there are no approved immunotherapies in this setting [7,8,9,10,11].

EOC cells overexpress several potentially targetable antigens minimally expressed in normal tissues, including Wilms’ Tumor 1 (WT1; 65% of patients), mucin 1 (MUC1; 90% of patients), MUC16 (85% of patients), New York esophageal squamous cell carcinoma 1 (NY-ESO-1; 40% of patients), and YKL-40 (76% of patients) [12,13,14,15]. Phase I trials have demonstrated the safe and effective induction of antibody responses to monovalent and polyvalent vaccines in various “stages” of the disease process in patients with EOC [16,17,18,19,20]. Unfortunately, these findings have not translated into clinical benefit. GOG-255, a randomized phase II trial in patients with recurrent ovarian cancer who are in clinical remission, investigated the survival benefit of a polyvalent vaccine conjugate of GM2-keyhole limpet hemocyanin (KLH), Globo-H-KLH, Tn-MUC1-32mer-KLH, and Thompson Freidreich antigen (TF)-KLH [16]. Although vaccination maintenance failed to prolong survival, 50% of patients did not experience an immunoglobulin (Ig)M response to the individual antigens in the vaccine; therefore, the study results were difficult to interpret. A large phase III randomized controlled trial of passive immunotherapy with abagovomab, an anti-idiotypic antibody against the CA-125 tumor antigen, also failed to demonstrate a survival benefit in patients with ovarian cancer [21]. In both studies, the authors concluded that future vaccine trials should leverage the concomitant or metachronous administration of vaccines with immune-modulating agents, such as checkpoint inhibitors, to potentiate immune responses to vaccines. Pre-clinical data have demonstrated that combining immune checkpoint inhibition and vaccines results in enhanced anti-tumor immune responses [22]. There are currently more than 80 ongoing vaccine clinical trials in ovarian cancer; however, few are assessing combinations with immune checkpoint inhibition [11]. The rationale for combination therapy with immune checkpoint inhibition is to create a favorable immunologic environment by attenuating immunosuppressive mechanisms, thereby encouraging the action of effector T cells generated by tumor-associated antigen vaccines. Nivolumab, in particular, has been studied as a companion immune checkpoint inhibitor in several clinical trials due to its tolerability, and its potential to enhance the durability of response to cancer vaccines [22].

WT1 represents an ideal target for tumor-directed immunotherapy in EOC, as it is an oncofetal antigen with expression in normal adult tissues limited to kidney, ovary, testis, spleen, and mesothelial lining but with overexpression in EOC cells [23,24,25]. Immunohistochemistry (IHC) has demonstrated high WT1 expression in serous ovarian tumors compared to other histologic subtypes [26]. Various peptide sequences from the WT1 antigen have been identified as immunogenic and capable of evoking sustained cytotoxic T-cell (CTL) responses that, in turn, target and kill WT1-expressing cancer cells [27]. Initial data with the WT1 peptide vaccine galinpepimut-S (GPS), which is a mixture of two native and two heteroclitic (mutated analogue) WT1 peptides, have been promising. GPS immunization in patients with malignant pleural mesothelioma demonstrated improved overall survival (OS) in patients who received the vaccine in the adjuvant setting after frontline standard tumor debulking with trimodality therapy compared to controls [28]. A phase II trial demonstrated superior median progression-free survival (PFS) in high-risk patients with multiple myeloma who received GPS plus lenalidomide compared to historical immunomodulator-alone therapy following successful autologous stem cell transplant; there was also a trend toward higher rates of hematologic complete response and very good partial response in the subgroup of patients with long-term, sustained high frequency of peripheral blood WT1-specific CD4+ T cells [29]. Moreover, GPS monotherapy has been used in the maintenance setting in patients with acute myeloid leukemia (AML) who successfully achieved first or second complete remission, resulting in good tolerance, high frequency of WT1-specific T-cell response, and preliminary evidence of antileukemic activity leading to prolonged OS over historical controls [30,31,32]. Finally, GPS is being studied in an ongoing phase III open-label, randomized clinical trial as maintenance therapy in patients with AML who have successfully achieved second complete remission versus best available therapy (NCT04229979).

We investigated the safety and immunogenicity of a WT1-targeting non–HLA-restricted heteroclitic tetravalent peptide vaccine (GPS) in combination with immune checkpoint inhibition (the anti–PD-1 [programmed cell death protein 1] nivolumab) in an open-label, non-comparative phase I trial of patients with WT1-positive EOC in second or third remission (NCT02737787). The primary endpoint was safety. Exploratory analyses included T-cell and serological analyses to investigate cellular and humoral immune responses; estimated 1-year PFS rate; and potential association of PFS with T-cell responses or IgG production.

## 2. Materials and Methods

### 2.1. Eligibility Criteria

Patients with histologically confirmed WT1-positive, recurrent, platinum-sensitive or platinum-resistant EOC in second or third complete clinical remission within 4 months of prior chemotherapy were eligible. Clinical remission was defined as a serum CA-125 level within normal limits and physical examination and computed tomography (CT) or magnetic resonance imaging (MRI) without objective evidence of disease. Patients were required to have a Karnofsky performance status of ≥70%. Patients with active infection requiring systemic treatments, those with known or suspected autoimmune disease, and those requiring systemic treatment with corticosteroids (>10 mg daily prednisone equivalents) or other immunosuppressive medications within 14 days of study drug administration were excluded. 

Screening for WT1 positivity was performed from archival fresh-frozen paraffin-embedded tissue, unstained slides, or fresh tissue by IHC as previously described [33], and patients harboring WT1-positive tumors (IHC score ≥6) were eligible for study enrollment.

This study was approved by the Memorial Sloan Kettering Cancer Center (MSK) Institutional Review Board (protocol #15-247). Patients signed informed consent for WT1 tumor testing and study treatment participation.

### 2.2. Treatment Plan

Patients received six doses of a vaccine containing 800 mcg of GPS (SELLAS Life Sciences Group, Inc., New York, NY, USA) administered subcutaneously (s.c.) at a 1:1 ratio with 0.7 mL of Montanide ISA 51 VG (an immune adjuvant containing a natural oil and refined emulsifier; SEPPIC, Inc., Fairfield, NJ, USA) over a 10-week period (weeks 0, 2, 4, 6, 8, and 10). Vaccines were administered with seven doses of the anti–PD-1 IgG4 monoclonal antibody nivolumab (Bristol Myers Squibb, Lawrenceville, NJ, USA) at 3 mg/kg intravenously (IV) over a 12-week period (weeks 0, 2, 4, 6, 8, 10, and 12). Injection sites were primed with 70 mcg sargramostim (human granulocyte-macrophage colony-stimulating factor [huGM-CSF]; Partner Therapeutics, Inc., Lexington, MA, USA) administered s.c. both 2 days before and on the day of each GPS vaccine. Vaccines were administered in patients’ extremities, and vaccination sites were rotated with each dose. Patients who remained in remission were offered a maintenance course of vaccination with GPS plus Montanide (after sargramostim priming)—without nivolumab—at weeks 19, 27, 35, and 43 (Appendix A). CT scans were obtained at baseline, week 15, and every 3 months thereafter for up to 1 year in those without disease progression. CA-125 levels were obtained at baseline, weeks 6, 15, and every 3 months thereafter for up to 1 year in those without disease progression. Treatment was discontinued at the end of the maintenance period, at the time of disease progression, or with a dose-limiting toxicity (DLT). 

### 2.3. Vaccine Preparation

To enhance the immunogenicity of the WT1 vaccine, synthetic analogue peptides were developed from WT1 protein sequences using computer prediction analysis. After substituting single or double amino acids at key HLA A*0201 binding positions, peptides were directly assayed for their ability to stabilize major histocompatibility (MHC) class I A*0201 molecules on the surface of a T2 cell line negative for expression of transporter associated with antigen processing (TAP). Avidly binding peptides were assayed in an antigen-specific T-cell expansion in vitro system, and their abilities to elicit HLA-restricted, peptide-specific CD8+ CTL responses were assessed using purified T cells from healthy donors. CD8+ T cells stimulated by the synthetic peptides demonstrated cross-reactivity with native WT1 peptides (a heteroclitic response) and the ability to kill HLA-matched chronic myelogenous leukemia (CML) blasts [34]. Two synthetic analogue peptides (WT1-A1 HLA class I peptide [9 amino acids] and WT1-122A1 long HLA class II peptide [9 amino acids]) generated a more effective immune response than native peptides; these were combined with two longer WT1 sequences capable of inducing in vitro CD4+ and CD8+ responses (namely, the WT1-427 long [19 amino acids] and WT1-331 long [22 amino acids] HLA class II peptides) to construct the tetravalent GPS vaccine [31]. Three peptides were designed to stimulate common HLA-antigen D related (HLA-DR) expressing cells, and one peptide was designed to stimulate HLA A*0201 cells. The provenance of the GPS vaccine has been previously described [22,35,36,37].

### 2.4. Toxicity Evaluation

Toxicity was graded by the National Cancer Institute Common Terminology Criteria for Adverse Events version 4.0 (CTCAE v4.0), and assessments were performed at baseline and weeks 1, 2, 4, 6, 8, 10, and 12, and 3 weeks after therapy completion at week 15. Thereafter, adverse events were reported every 3 months for up to 1 year or until disease progression. DLTs were evaluated within 30 days of first vaccination, and included any grade 5 adverse event, any grade 2 drug-related ophthalmic adverse event, any grade 3 drug-related adverse event or laboratory abnormality lasting >72 h (exceptions detailed in the study protocol), any grade 4 drug-related adverse event or laboratory abnormality (exceptions detailed in the study protocol), grade 3 injection site reaction, grade 3 fever, or any adverse event presenting a substantial clinical risk to the patient as judged by the investigator. The detection of more than two DLTs among 10 treated patients was grounds for study discontinuation due to safety concerns.

### 2.5. Immune Response Evaluation

Immune monitoring occurred at screening, weeks 0, 6, 15, and every 3 months until recurrence or study end. 

To investigate antigen-specific T-cell responses, cytokine production after antigenic stimulation was used as a readout via intracellular cytokine staining with a T-cell flow cytometry panel on thawed peripheral blood mononuclear cell (PBMC) samples cultured with the relevant vaccine antigens. After initial thaw, cells were rested overnight at 37 °C and then separately stimulated with six different peptide antigens (WT1-A and WT1-A1 in HLA-A*02 patients, as well as 122-A, 122-A1, 331 L, and 427 L) or a CEF-positive control peptide pool consisting of 32 MHC class-I restricted viral peptides from human cytomegalovirus, Epstein-Barr virus, and influenza virus, along with supplemental interleukin (IL)-2 and IL-15, which were replaced every 2–3 days. At the end of the 10-day culture period, cells were counted, washed, and restimulated with the same initial peptide antigens (or an irrelevant long peptide B2A2 from the BCR-ABL fusion protein) plus fluorescence-labeled antibody to degranulation marker CD107a (H4A3, BioLegend) for 6 h, with the last 4 h in the presence of Golgi transport inhibitors Brefeldin-A and monensin. The cells stimulated with irrelevant peptide were used as negative controls to establish cytokine positivity gates in flow. After the restimulation period, cells were then stained with a viability dye and antibodies to surface CD8 (SK1, BD Biosciences), fixed, permeabilized, washed, and refrigerated overnight in permeabilization buffer. On the following day, cells were stained with fluorescence-labeled antibodies to CD3 (UCHT1, BD Biosciences), CD4 (SK3, BD Biosciences), and the following intracellular cytokines: interferon (IFN)-γ (B27, BD Biosciences), IL-2 (5344.111, BD Biosciences), and tumor necrosis factor (TNF)-α (Mab11, BD Biosciences). Cells were then washed and resuspended in phosphate-buffered saline (PBS)/1% fetal bovine serum (FBS) buffer for acquisition on a BD Biosciences Fortessa X-20 flow cytometer. After acquisition, flow cytometry data were analyzed using FlowJo software (FlowJo, LLC) to assess polyfunctional T-cell responses in gated CD4 and CD8 T-cell subsets, with TNF-α production used as the representative cytokine for measurement of the antigen-specific T-cell response given its relative prominence across the various cytokines evaluated. Percentages of cytokine-positive T cells with antigen responses 2-fold higher compared to control (irrelevant) peptides were considered positive immune responses. In addition to the intracellular cytokine staining panel, a separate T-cell flow immunophenotyping panel was used to examine CD4+ and CD8+ T-cell expression of CD38 (HIT2, BioLegend), PD-1 (MIH4, BD Biosciences), programmed death-ligand 1 (PD-L1), Ki67 (B56, BD Biosciences), Granzyme B (GB11, BD Biosciences), and CD25 (M-A251, BD Biosciences). For these assays, isotype controls were used for each marker to set the gates for determining positivity. 

To investigate humoral immune responses, IgG was measured by enzyme-linked immunosorbent assay (ELISA) against individual WT1 peptides within GPS as well as full-length WT1 and control antigens NY-ESO-1, MUC16, and DHFR. Measurements were conducted by ELISA at baseline (screening and week 0), weeks 6, 15, 21–27, and 48 (or at time of disease progression for patients coming off study early). 

### 2.6. Statistical Considerations

The primary endpoint of this phase I study was the safety of nivolumab in combination with the GPS vaccine (administered as an emulsion with Montanide and using sargramostim as an immune adjuvant/pre-vaccination “primer”). The initial plan was to enroll 10 evaluable patients, i.e., those who completed at least one dose of GPS vaccination. Detection of 2 or fewer DLTs would deem the combination safe. Descriptive characteristics were used to describe toxicities and immune responses. 

For secondary objectives, the antigen-specific T-cell response was considered positive if values were at least 2-fold higher with test peptides compared to the background observed with control (irrelevant) peptides. Criteria for IgG immune responders were based on prior studies and defined as patients with anti-WT1 antibody titers that increased from undetectable to ≥1:40 post-treatment, or those with an ≥8-fold increase over detectable pretreatment levels at any time point [19]. PFS was measured from the start date of the preceding chemotherapy session to the date of progression of disease by Response Evaluation Criteria in Solid Tumors (RECIST) 1.1 criteria or death (while on study). This interval was chosen following recommendations for maintenance trials in ovarian cancer to measure the effect of consolidation therapy independent of the effect of the therapy that achieved clinical remission [38]. Patients alive and progression free were censored at the time of their last available/recorded follow-up, and 1-year PFS was defined as an endpoint at the initiation of the study. 

## 3. Results

### 3.1. Patient Characteristics

Twenty-five patients consented for WT1 testing of their tumor (MSK protocol #15-247; NCT02737787) between June 2016 and July 2017. Of 25 patients, 23 were tested, and 22 (96%) of 23 had WT1-positive tumors. During screening, 10 patients were deemed ineligible due to persistent disease on imaging, and 1 withdrew consent. Eleven patients consented to study treatment. Baseline demographics and disease characteristics are presented in Table 1. All patients had a normal CA-125 level and did not have radiographic evidence of disease at the time of study enrollment. The median CA-125 level at enrollment was 7 units/mL (range, 3–21; normal <35 units/mL).

The median patient age was 62 years (range, 40–74), and all patients had a Karnofsky Performance Status of 90% or ECOG 0. The most common HLA-A subtypes were HLA-A*0201 (36%), HLA-A*0101 (18%), and HLA-A*2402 (18%) (Appendix A). The median WT1 score was 12 (range, 8–12). Ten patients had high-grade serous carcinoma, and one had low-grade serous carcinoma. Nine (82%) had International Federation of Gynecology and Obstetrics (FIGO) stage III disease at diagnosis. All patients had platinum-sensitive disease, and the median number of prior chemotherapy regimens was two (range, 1–4). The median time from last dose of cytotoxic therapy to initiation of the trial intervention was 1.6 months (range, 0.7–2.6). 

### 3.2. Safety and Tolerability

All 11 patients were included in the safety analysis. The most common adverse event was injection site reaction (64%; all grade 1); 36% had arthralgia (grade 1 and 2); and 36% experienced fatigue (grade 1 and 2). In total, there were 25 grade 1 toxicities and 8 grade 2 toxicities. One patient experienced a grade 3 immune-related myositis/myocarditis attributed to the study intervention, which was considered a DLT. There were no grade 4 or 5 toxicities. Eight (73%) of the eleven patients reported at least one adverse event and seven patients (64%) had more than one adverse event. Data on maximum toxicity for treatment-related adverse events are reported in Table 2.

The DLT occurred in a 75-year-old woman (HLA-A*0101) with a history of multivessel coronary artery disease, hypertension, hyperlipidemia, obesity, and diet-controlled type 2 diabetes mellitus. After two doses of nivolumab 3 mg/kg and GPS (21 days after study initiation), she experienced complete heart block. Laboratory findings were notable for elevated troponin I at 17.1 ng/mL (normal <0.6 ng/mL) and CK-MB at 249.4 ng/dL (normal <5.0 ng/dL). Cardiac catheterization demonstrated no new coronary artery disease, and transthoracic echocardiography showed no ischemic cardiomyopathy or valvular disease. The patient required pacemaker insertion. Based on clinical, laboratory, and imaging findings, the patient met the criteria for clinically suspected myocarditis [39,40]. A week later, she developed diffuse myositis and dysphagia. Muscle biopsy revealed focal myofiber necrosis, mild inflammation, and a type II myofiber atrophy consistent with an immune-mediated myopathic disorder. The immune-related adverse event was attributed to nivolumab, and the patient required IV methylprednisolone and insertion of a feeding tube. This was followed by a prolonged steroid taper and long-term rehabilitation including physical therapy. The patient thereafter remained under the care of a cardiologist for ongoing management of her permanent pacemaker. Six months after her discontinuation from the study, the patient returned to her baseline status and subsequently resumed chemotherapy with her medical oncologist who continued to follow her closely.

### 3.3. Immune Response

Investigations of peripheral blood immune cell subpopulation frequencies and surface activation markers were conducted at screening, throughout treatment, and at 3–6 months of follow-up. For antigen-specific T-cell response assays, 51 samples from 11 patients were available. For immunophenotyping assays, 35 samples from 8 patients were available.

We first assessed frequencies of peripheral blood CD4 and CD8 T-cell subsets, as well as their phenotypic expression of PD-1, PD-L1, CD38, CD25, Ki67, and granzyme B. Among the patients with available samples, the majority did not have significant changes in circulating CD4 or CD8 T-cell populations (Figure 1). There was a trend toward decreased expression of PD-1 post-treatment in CD4+ and CD8+ T-cell subsets in the majority of samples. There were no consistent trends in phenotypic expression of CD38, CD25, Ki67, or granzyme B by CD4 or CD8 T cells among the study population (Appendix A).

We then examined cytokine responses (TNFα, IFNγ, and IL-2) of CD4 and CD8 T cells to test peptides. Ten (91%) of eleven patients experienced vaccine-specific T-cell responses, primarily within CD4+ T cells (Figure 2). These responses peaked at 6–15 weeks in several patients. The most robust responses against specific peptides were CD4+ T-cell responses against 331 L and 427 L peptides, with weaker responses against 122-A and 122-A1 in some patients (Figure 2 and Appendix A). The longevity of these responses was limited; only two out of six patients exhibited responses at longer-term follow-up visits. 

To assess humoral responses, we measured serum levels of IgG against individual WT1 peptides within the GPS vaccine and against the full-length WT1 protein at screening, prior to the first dose of vaccine, and throughout each patient’s treatment course (range, 0–48 weeks) (Figure 3). Among eight evaluable patients, increased serum levels of IgG were induced for 88% between weeks 6–27 for both WT1 peptides and the full-length WT1 protein. The induction of WT1-specific IgG antibodies following GPS administration was highly consistent among patients, some detectable as early as week 6, and achieving high titers (>1/10,000) in most patients. Induction patterns for IgG antibodies against individual peptides within the GPS mixture were similar to those against the full-length WT1 protein.

### 3.4. Exploratory Analysis

The 1-year PFS rate was 64% in the intent-to-treat group (n = 11) and 70% in the group of patients who received >2 doses of GPS and nivolumab (n = 10). CA-125 results are depicted in Appendix A. Among all 11 patients, the median PFS was 12.9 months (95% CI, 2.8–23.1) (Figure 4). Two patients, both with three prior lines of cytotoxic therapy for high-grade serous carcinoma, remained recurrence-free at 55.3 months and 53.8 months, respectively. One patient had a WT1 score of 12, while the other patient had a WT1 score of 8. Both patients displayed CD4+ T-cell responses and IgG responses against individual WT1 peptides; their HLA subtypes were HLA A*0301 and HLA A*2402, respectively.

## 4. Discussion

In this phase I trial, the combination of GPS vaccine and nivolumab was deemed safe. The most common adverse event was localized injection site reaction in 64% of patients. 

This common adverse event was similarly reported in 85% of patients in a study of GPS in malignant pleural mesothelioma [25]. A large systematic review reported injection site reactions in 67.9% of patients receiving the heavy depot oil Montanide, in which the GPS peptide mixture is emulsified [41]. 

One patient experienced a DLT, myocarditis/myositis attributed to nivolumab, which was classified as a grade 3 adverse event. Activation of a non-target immune response in the heart and skeletal muscle is a rare but previously described adverse event of immune checkpoint inhibition [42,43]. This phenomenon is attributed to the expression of certain receptors, including PD-1, on cardiomyocytes and myocytes [43]. Multi-institutional retrospective studies and safety databases have reported immune checkpoint inhibition-associated myocarditis rates of 0.06–1.14% and grade 3–4 myositis rates of 0.15–0.24% [39,44]. This frequency indicates these are rare but potentially severe complications of immune checkpoint inhibition [45]. Within the GPS vaccine literature, there are no known reports of myocarditis, suggesting that this presentation was most likely uniquely attributable to the immune checkpoint inhibition.

Myocarditis has the highest fatality rate among immune checkpoint inhibition toxic events, reported at 39.7% in one large retrospective study [45]. Although baseline cardiac disease has not been associated with an increased risk of immune checkpoint inhibition-associated myocarditis, our patient had a medical history notable for diabetes mellitus and obesity, which are risk factors for this rare complication [39]. Furthermore, her presentation with myocarditis at 21 days following initiation of nivolumab is consistent with the median time of onset reported in the literature at 27–34 days (range, 5–155) [39,46]. Fortunately, the discontinuation of immune checkpoint inhibition, pacemaker insertion, and treatment with IV steroids prevented an immune checkpoint inhibition-associated fatality. 

Assessment of immune response, including frequencies of antigen-specific CD4 and CD8 T cells over sequential vaccination, has been a key exploratory endpoint of GPS trials [28,29,30,31,32]. The ability to correlate multiple parameters, including cytokine channel, specific WT1 peptides, and CD4 and CD8 T-cell responses has been limited. In this study, 10 (91%) of 11 patients experienced vaccine-specific T-cell responses, primarily within CD4+ T cells but also within CD8+ T-cell subpopulations. As in other studies, observed responses peaked at 6–15 weeks, consistent with the cumulative effect [34,37,47]. Further, IgG responses to an aggregate of WT1 peptides within GPS and the full-length WT1 protein were induced in 88% of evaluable patients. These findings support the generation of the WT1-specific immune response in the majority of patients, and suggest that combining GPS and nivolumab to bypass local immune tolerance and negative signaling can rapidly induce T-cell responses in patients with WT1-expressing EOC. As WT-1 positivity is present in approximately 65% of patients with WT1-positive EOC, there is the potential for this vaccine and immune checkpoint inhibition combination to effectively generate an immune response in a large percentage of patients with EOC [12]. The therapeutic implications of this immune response warrant additional investigation. Notably, the decrease in PD-1-positive CD4 and CD8 T cells may be due to a technical artifact caused by interference of anti–PD-1 antibody flow staining by bound therapeutic nivolumab; this would need to be resolved in future experiments. Additionally, we were unable to disentangle antibody responses and provide additional clarity on whether responses differed by WT1 peptide, as IgG responses were measured in aggregate against the WT1 peptide fragment pool. While the generation of a humoral immune response complemented the cellular immune response findings with implications for durability of response, future investigations could examine immune responses to a single epitope.

As an exploratory objective, we assessed 1-year PFS. The 64% PFS rate in the intent-to-treat group and approximately 70% rate in patients who received >2 doses of the GPS and nivolumab combination compare favorably to historic rates of up to 55% in comparable patient populations [38]. The prolonged disease-free interval in two patients, who are in remission >50 months, is notable. Other studies on immunomodulation in ovarian cancer, such as the NINJA and NRG GY003 trials, have also reported extended duration of response in patients who respond to immune checkpoint inhibition [10,48]. These data suggest that durable responses to immunotherapy are attainable in certain patients with WT1-positive EOC, in contrast to chemotherapy, for which responses in the recurrent setting are typically short-lived. 

The limitations of this study are primarily related to the small study size. Primarily, outcomes including progression-free survival were difficult to correlate with immune response data given the small sample size. In fact, the majority of patients had cellular and humoral response signals without obvious correlates to survival outcomes, although the population was preselected for patients with WT1-positive EOC. All patients were in remission at the time of study entry, and therefore a randomized trial would be required to ascertain the relative contribution of the study interventions on survival outcomes. Certainly, correlating the translational data to patient outcomes in order to better select candidates for this therapy would be beneficial in moving forward. 

## 5. Conclusions

This phase I, open-label, non-randomized trial of combination therapy with an anti–PD-1 (nivolumab) and GPS, a novel WT1-targeting non–HLA-restricted heteroclitic tetravalent peptide vaccine, administered as maintenance therapy in patients with WT1-positive ovarian cancer in second or third remission met its predetermined endpoint of safety and tolerability. Preliminary data have shown WT1-specific immune response and encouraging PFS results. These findings have served as the basis for an ongoing phase I/II study of GPS in combination with the anti–PD-1 drug pembrolizumab in patients with ovarian cancer and other solid tumors (SLS17-201/MK3475-770; NCT03761914).

## Figures and Tables

**Figure 1 cancers-15-01458-f001:**
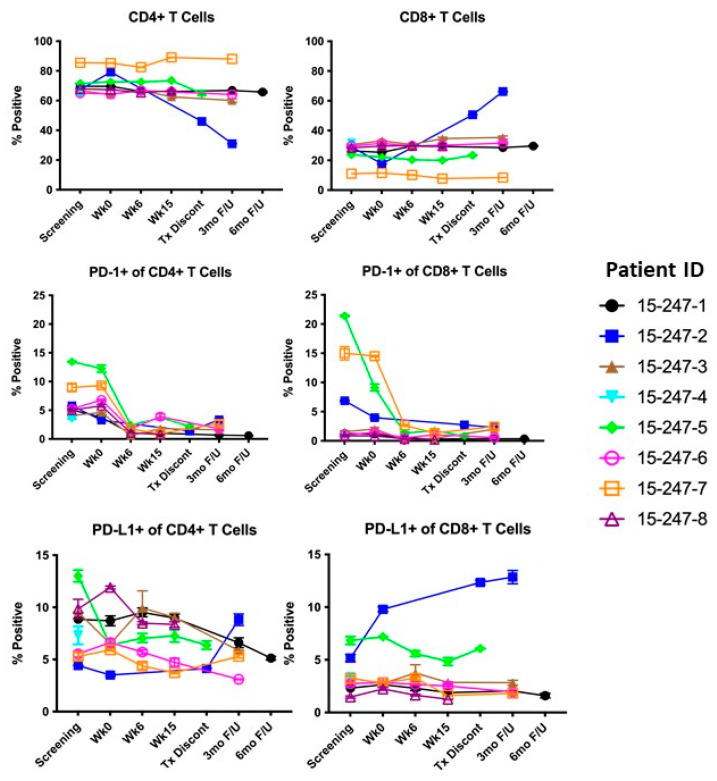
Summary of cell subpopulation frequencies over time. Tx Dicont, Treatment Discontinuation; 3 mo F/U, 3-month follow-up; 6 mo F/U, 6-month follow-up; PD-1, programmed cell death protein 1; PD-L1, programmed death-ligand 1.

**Figure 2 cancers-15-01458-f002:**
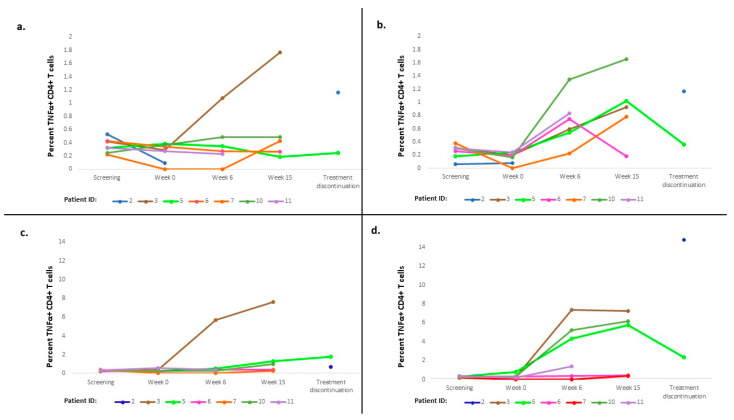
Cytokine responses of CD4 T cells to test peptides. (**a**) CD4 immune response against 122A in non-HLA2 patients (TNFα); (**b**) CD4 immune response against 122A1 in non-HLA2 patients (TNFα); (**c**) CD4 immune response against 427 L in non-HLA2 patients (TNFα); (**d**) CD4 immune response against 331 L in non-HLA2 patients (TNFα). HLA, human leukocyte antigen; TNF, tumor necrosis factor.

**Figure 3 cancers-15-01458-f003:**
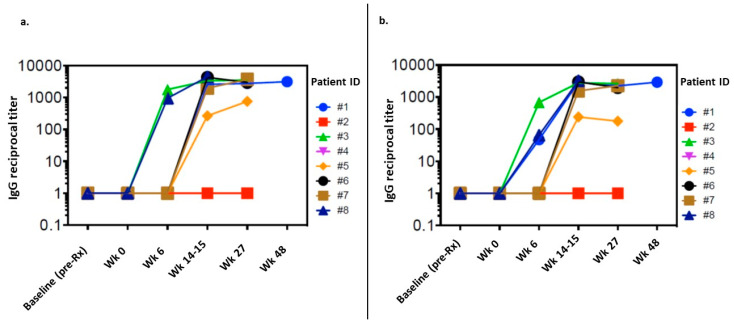
Serum levels of IgG against (**a**) a WT1 peptide pool from the GPS vaccine; (**b**) against the full-length WT1 protein. Ig, immunoglobulin; WT1, Wilms’ Tumor 1; GPS, galinpepimut-S.

**Figure 4 cancers-15-01458-f004:**
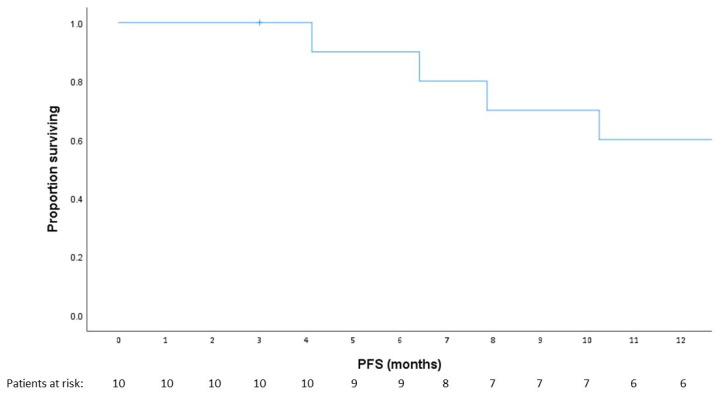
Progression-free survival (PFS). Kaplan–Meier curve demonstrating PFS in patients on trial.

**Table 1 cancers-15-01458-t001:** Study population (N = 11).

Patient Characteristic	n (%)
**Median age, years (range)**	62 (40–74)
**Median CA-125 level at study enrollment, units/mL (range)**	7 (3–21)
**Race**	
Asian	1 (9%)
Black	1 (9%)
White	9 (82%)
**HLA subtype**	
A*0201	4 (36%)
A*0101	2 (18%)
A*2402	2 (18%)
A*1101	1 (9%)
A*0301	1 (9%)
A*23	1 (9%)
**Histologic subtype**	
High-grade serous	10 (91%)
Low-grade serous	1 (9%)
**Stage at diagnosis**	
II	1 (9%)
III	9 (82%)
IV	1 (9%)
**Prior lines of chemotherapy**	
1	1 (9%)
2	5 (46%)
3	4 (36%)
4	1 (9%)

**Table 2 cancers-15-01458-t002:** Patients per maximum toxicity for treatment-related events (N = 11).

Treatment-Related Adverse Event	Grade 1 n (%)	Grade 2 n (%)	Grade 3 n (%)
Injection site reaction	7 (64%)	0	0
Arthralgia	3 (27%)	1 (9%)	0
Fatigue	3 (27%)	1 (9%)	0
Myositis/myocarditis	0	0	1 (9%)
Rash	1 (9%)	1 (9%)	0
Decreased white blood cell counts	1 (9%)	1 (9%)	0
Decreased platelet counts	0	1 (9%)	0
Hypothyroidism	0	1 (9%)	0
Pneumonitis	1 (9%)	0	0
Alopecia	0	1 (9%)	0
Diarrhea	1 (9%)	0	0
Vision changes (floaters)	1 (9%)	0	0
Pruritis	1 (9%)	0	0
Weight loss	0	1 (9%)	0

## Data Availability

The data presented in this study are available upon reasonable request from the corresponding author in accordance with institutional guidelines. The data are not publicly available due to privacy restrictions.

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
