# Peer review of "Phase I Study of a Multivalent WT1 Peptide Vaccine (Galinpepimut-S) in Combination with Nivolumab in Patients with WT1-Expressing Ovarian Cancer in Second or Third Remission"

_cancers, 2023, doi:10.3390/cancers15051458_

Round 1
Reviewer 1 Report
This article discusses a novel immunotherapy option to treat epithelial ovarian cancer in the form of a peptide vaccine targeting the Wilms’ Tumor 1 (WT1) protein, that is found overexpressed in this type of tumor, in conjunction with the anti-PD-1 nivolumab and a phase I trial of patients with WT1-positive epithelial ovarian cancer in second or third remission. According to the authors, the targeting of WT1 occurs by exploiting the immunogenic properties of sequences of the WT1 antigen capable of initiating cytotoxic T-cell responses in the form of a mixture of WT1 peptides in the vaccine, which is referred to as galinpepimut-S (GPS). The authors indicate that the results of this phase I clinical trial show that the combination of GPS vaccine and nivolumab was a relatively safe and effective option for the treatment of WT1-positive epithelial ovarian cancer, with the generation of WT1-specific immune response in the majority of patients in the trial, which, as the authors point out, makes this new combination worth investigating more. The authors suggest further aspects that require investigation. The GPS vaccine proteins were chosen with care to be able to generate a more effective immune response compared to native peptides and capable of inducing CD4+ and CD8+ responses. Evaluation of the immune response involved investigating cytokine production after antigenic stimulation of patient-derived cells in vitro, as well as immune marker expression and humoral immune response via IgG measurement. Overall, I believe the authors demonstrated to an acceptable level the immune response to the combination treatment and qualified the toxicity level of the treatment and explained well the results obtained.
Below are a few minor comments:
1. Could the authors further clarify why the maintenance courses of GPS vaccination for patients are still in remission?
2. According to the Eligibility criteria, clinical remission was defined as a serum CA-125 level/ CT/ MRI. Still, the results of these judgments are not presented in the paper (including clinical remission and clinical non-remission).
3. The peptides were developed from WT1 protein sequences using computer prediction analysis. So, I would like to ask if the authors have done relevant tests at the cellular level or if any relevant studies have been done?
4. If possible -- it would be nice to add a treatment timeline diagram so that it will be easier to understand by the reader.
Author Response
This article discusses a novel immunotherapy option to treat epithelial ovarian cancer in the form of a peptide vaccine targeting the Wilms’ Tumor 1 (WT1) protein, that is found overexpressed in this type of tumor, in conjunction with the anti-PD-1 nivolumab and a phase I trial of patients with WT1-positive epithelial ovarian cancer in second or third remission. According to the authors, the targeting of WT1 occurs by exploiting the immunogenic properties of sequences of the WT1 antigen capable of initiating cytotoxic T-cell responses in the form of a mixture of WT1 peptides in the vaccine, which is referred to as galinpepimut-S (GPS). The authors indicate that the results of this phase I clinical trial show that the combination of GPS vaccine and nivolumab was a relatively safe and effective option for the treatment of WT1-positive epithelial ovarian cancer, with the generation of WT1-specific immune response in the majority of patients in the trial, which, as the authors point out, makes this new combination worth investigating more. The authors suggest further aspects that require investigation. The GPS vaccine proteins were chosen with care to be able to generate a more effective immune response compared to native peptides and capable of inducing CD4+ and CD8+ responses. Evaluation of the immune response involved investigating cytokine production after antigenic stimulation of patient-derived cells in vitro, as well as immune marker expression and humoral immune response via IgG measurement. Overall, I believe the authors demonstrated to an acceptable level the immune response to the combination treatment and qualified the toxicity level of the treatment and explained well the results obtained.
Response: We greatly appreciate the reviewer's comments.
Below are a few minor comments:
- Could the authors further clarify why the maintenance courses of GPS vaccination for patients are still in remission?
Response: Yes. All patients were in remission at study entry. The GPS vaccine was well tolerated in prior studies and the rationale for offering the maintenance GPS vaccination phase for those in ongoing remission was to provide patients with additional intermittent cancer associated antigen (WT1) stimulation that would be relatively well tolerated following the prior immune priming phase of the combination of nivolumab (immune-checkpoint inhibition) and the GPS vaccine.
2. According to the Eligibility criteria, clinical remission was defined as a serum CA-125 level/ CT/ MRI. Still, the results of these judgments are not presented in the paper (including clinical remission and clinical non-remission).
Response: We agree with the reviewer that this was not clearly presented. We have now added this information at line 268 in the results section as well as to Table 1. The attached graph of CA125 values is now included as a supplementary figure 9. All patients had a normal CA-125 (<35 at our institution) and imaging was also consistent with remission.

3. The peptides were developed from WT1 protein sequences using computer prediction analysis. So, I would like to ask if the authors have done relevant tests at the cellular level or if any relevant studies have been done?
Response - Yes we have. Please find below a comprehensive summary of the relevant studies. The figures that accompany this are included in the PDF that is attached.
The list of key papers supporting the Discovery and Preclinical Biology of GPS as a cancer immunotherapeutic agent follows:
a. The seminal paper for choosing WT1 as the preferred antigenic target of vaccines for cancer immunotherapy is the one by Dr. Mac Cheever and collaborators, as per the citation below:
- Cheever MA, Allison JP, Ferris AS, et al. The prioritization of cancer antigens: a national cancer institute pilot project for the acceleration of translational research. Clin Cancer Res. 2009;15(17): 5323-37.
b. The rationale for the development of WT1 vaccines in myeloid malignancies is summarized in the publication below:
- Dao T, Scheinberg DA. Peptide vaccines for myeloid leukaemias. Best Pract Res Clin Haematol. 2008 Sep;21(3):391-404.
The key preclinical studies on GPS have been been published previously.
A summary of the findings follows (for each of the 4 GPS constituent peptides).
WT1-A1 heteroclitic 9-mer peptide with sequence: Y*MFPNAPYL
(*:R>Y substitution in the heteroclitic Y-containing version vs. the native sequences).
Peptide WT1-A1: HLA-A2 Binding
In vitro, the modified A1 peptide is more effective in causing human T-cells to recognize WT1-bearing targets than its native sequence. See Fig. 1.
Fig. 1. (see attached PDF that has this figure as unable to paste here)
Abbreviations: CD=cluster of differentiation; HLA=human leukocyte antigen; NaCl= sodium chloride; R=arginine; Y=tyrosine.
Left: The R1Y variant binds more tightly to HLA-A2 than the native peptide as shown by the greater thermal stability of the R1Y/HLA-A2 complex. The apparent Tm of each complex is indicated on the curves. The NaCl concentration in this experiment was 75 mM.
Right: The difference in thermal stability between the R1Y and native complexes diminishes as the ionic strength of the solution is increased, revealing the role that electrostatics plays in enhancing peptide binding.
Peptide WT1-A1: CD8+/CD3+ Responses From HLA‑A*0201 Donors
As illustrated in Fig. 2, after 2 rounds of antigen stimulation, the modified A1 peptide generated a robust immune response, but the native sequence (A) remained near the limit of detection by IFN-γ enzyme-linked immunospot (ELISPOT) testing.
Fig. 2. (see attached PDF that has this figure)
Abbreviations: APC=antigen presenting cell; ELISPOT=enzyme-linked immunospot assay; HLA=human leukocyte antigen; IFN‑γ=interferon gamma; WT1=Wilms’ tumor 1 antigen.
T-cells were stimulated in vitro with the peptides A and A1. Black bars: CD8+ plus APC T2; gray bars: CD8+ plus T2 pulsed with analog peptide; light gray bars: CD8+ plus T2 pulsed with native peptide; white bars: CD8+ plus T2 pulsed with negative peptide. Results: T‑cells after 2 rounds of stimulation with A or A1.
The Y axis represents the number of spots per 1_105 CD8+/CD3+ T-cells. The X axis shows the different peptides used for stimulations. Experiments were performed in triplicate and confirmed 3 to 5 times.
Source: Ibarz et al 2006.
Peptide 122A1-Long
GPS 122A1 Long: 122 (native) and 122 A1 (heteroclitic) Stimulate CD8+ T-cell Responses
Once amino acid 126 was mutated from an R to a Y in the 122A1 Long galinpepimut-S drug substance, a heteroclitic WT1-A1 peptide (CD8+-activating) was effectively embedded within a long CD4 epitope. The peptide was extended by an additional number of amino acids at the N and C termini to increase the likelihood of better processing and to make the epitope more promiscuous. Data indicated the 122A1 peptide was properly processed and presented by APCs and could stimulate CD8 and CD4 T-cells which recognize native WT1 antigen (Fig. 3) (May et al 2007). Because this peptide contained a CD8 epitope ‘buried’ inside a CD4 epitope, the CTL response was expected to be more robust than its native form.
Fig. 3. (see attached PDF)
Table 1 Galinpepimut-S A1 and 122A1 Long Drug Substances: In Vitro Pharmacology
|
Peptide Tested |
Experiment |
Outcome(s) |
Reference |
|
WT1-A1 |
IFN-γ ELISPOT Using an optimized T-cell expansion system, with monocyte-derived DCs, CD14+ APCs, and purified CD3+ T-cells, the ability of the synthetic analog to stimulate peptide-specific CTL was tested. Ten healthy HLA-A*0201 donors were studied. Cells were stimulated in vitro, and secreted IFN-γ when challenged. 51CR release assay T‑cell lines and clones obtained after multiple stimulations with the modified WT1 126 peptide, to assess functional killing. |
After 2 rounds of APC stimulation, the modified peptide was able to generate a robust immune response, whereas the native sequence was close to the limit of detection. T-cells generated in vitro in the presence of the modified WT1 126 peptide were able to kill T2 leukemia cells pulsed with specific peptides, but not T2 cells without peptide or T2 cells pulsed with a control peptide. T-cells stimulated with the modified peptide recognized the native sequences as well. |
Error! Reference source Pinilla-Ibarz et al 2006 |
|
122A1-L (Modified) |
Cell stimulation assays CD3+ and CD4+ cells from healthy donors were stimulated with autologous DCs that had been loaded with peptide. Induction of the immune response was determined by an IFN-γ assay. |
Modified peptide WT1 122 induced a stronger T-cell response than native WT1 122, as T-cells recognized both 122 and 122 A1. Further testing with each peptide against donor cells with different HLA-DRB-1 types produced wide variation in terms of predicted binding scores. |
May et al 2007 |
|
122A1-L (Modified) |
Cross-priming Tumor lysates were prepared from 3 cell lines, an e1a2 leukemia line, a biphasic mesothelioma line, and a malignant melanoma line, and were incubated with cells from healthy A*0201+ donors. These cells were then used to stimulate autologous CD3+ T-cells. Stimulated T-cells were then tested against autologous DCs with the WT1 122 (modified) peptide. The experiment was then reversed. |
Those T-cells stimulated with WT1+ lysates recognized HLA class II peptides but those stimulated with DCs pulsed with WT1- lysates did not. T-cells stimulated with WT1+ lysates were only able to recognize modified peptide WT1 122 in 3 of 5 experiments. WT1-DR peptide-stimulated T-cells were able to recognize mesothelioma tumor cells, indicating quantities of WT1 peptides on tumor cells to induce a response. |
May et al 2007 |
|
122A1-L (Modified) |
Cytotoxicity assays CD3+ T-cells from donors were stimulated with peptide WT1 122, and were then challenged with autologous CD14+ cells in the presence of several peptides, or with a melanoma cell line in IFN‑γ ELISPOT assays. CD8+ T-cells were then isolated and used as effector cells in a 51Cr release assay. |
CD8+ and CD4+ cells were able to be activated by modified peptide WT1 122 against WT1. Modified peptide WT1 122 was a stronger stimulator and hence was able to induce a cytotoxic response to WT1+ leukemia cells. The same CD8+ T-cells did not, however, recognize a WT1- lymphoma, except when it had been pulsed with modified peptide WT1 122. Repeat experimentation demonstrated that modified peptide WT1 122’s potency was variable in different donors. CD8+ T-cells stimulated with modified peptide WT1 122 were toxic to the mesothelioma cell line but not to the melanoma cell line; CD4+ cells were not cytotoxic to either cell line. |
May et al 2007 |
Abbreviations: DC=dendritic cell; 51Cr=chromium 51; APC=antigen presenting cell; ELISPOT=enzyme-linked immunospot; HLA=human leukocyte antigen; IFN‑γ=interferon gamma; WT1=Wilms’ tumor 1.
WT1-331 Long and WT1-427 Long Peptides
Peptide WT1-331 had been previously identified (Müller et al 2003), and the peptide contained in galinpepimut-S was modified (L for ‘Long’) by adding 3 amino acids to the N-terminal, and 4 amino acids to the C-terminus. This was performed for several reasons. Flanking residues outside the minimal epitope have been shown to greatly influence processing of the peptide (Eisenlohr al 1992). Additionally, MHC class II molecules have a more permissive binding pocket; therefore, an elongated peptide could potentially be a more promiscuous epitope, in that, it could bind to more than one MHC II molecule.
Peptide WT1-427 L (‘Long’) is a novel WT1 CD4+ epitope and its immunogenicity has been confirmed. Processing experiments have confirmed this peptide is processed and presented by APCs when treated with either WT1-expressing CML or mesothelioma tumor lysates.
The physicochemical properties of heteroclitic (mutated) peptides containing the WT1-A1 sequence (one of the peptides contained in the GPS mixture) specifically with regard to HLA binding was also independently studied by two 3rd party laboratories by teams led by Dr. Graham Pawelec at the University of Tuebingen (Germany) and Dr. Brian Baker at Notre Dame University (Indiana, USA), respectively. These studies have been published, as cited below:
- Mueller L, Knights A, Pawelec G. Synthetic peptides derived from the Wilms’ tumor 1 protein sensitize human T lymphocytes to recognize chronic myelogenous leukemia cells. Hematol J. 2003;4:57-66.
- Borbulevych OY, Do P, Baker BM. Structures of native and affinity-enhanced WT1 epitopes bound to HLA-A*0201: implications for WT1-based cancer therapeutics. Mol Immunol. 2010;47(15): 2519–2524.
Also see:
- Eisenlohr LC, Yewdell JW, Bennink JR. Flanking sequences influence the presentation of an endogenously synthesized peptide to cytotoxic T lymphocytes. J Exp Med. 1992 Feb 1;175(2):481-7.
The in vitro immunobiology experiments were conducted at the Memorial Sloan Kettering Cancer Center (MSKCC) by Dr. David Scheinberg, a renowned expert in basic and translational research in peptide vaccine technology, at the time Head of the Leukemia Dept. at MSK and currently Head of the Dept. of Experimental Therapeutics at the same institution.
The pertinent references are as follows:
- Gomez-Nunez M, Pinilla-Ibarz J, Dao T, et al. Peptide binding motif predictive algorithms correspond with experimental binding of leukemia vaccine candidate peptides to HLA-A*0201 molecules. Leuk Res. 2006;30:1293–98.
- Pinilla-Ibarz J, May RJ, Korontsvit T, et al. Improved human T cell responses against synthetic HLA-0201 analog peptides derived from the WT1 oncoprotein. Leukemia. 2006;20:2025-33.
- May RJ, Dao T, Pinilla-Ibarz J, et al. Peptide epitopes from the Wilms' tumor 1 oncoprotein stimulate CD4+ and CD8+ T cells that recognize and kill human malignant mesothelioma tumor cells. Clin Cancer Res. 2007;13(15 Pt 1):4547-55.
- Gómez-Nuñez M, Haro KJ, Dao T, Chau D, Won A, Escobar-Alvarez S, Zakhaleva V, Korontsvit T, Gin DY, Scheinberg DA. Non-natural and photo-reactive amino acids as biochemical probes of immune function. PLoS One. 2008;3(12):e3938.
4. If possible -- it would be nice to add a treatment timeline diagram so that it will be easier to understand by the reader.
Agree - This was added as a new supplementary figure 8.

Reviewer 2 Report
This is a very well-written study. This phase I study aimed to assess the safety of a WT1 peptide vaccine in combination with nivolumab in patients with WT1-expressing ovarian cancer in second or third remissions.
Introduction: Very well written. No major comments.
Methodology: Well explained. With a detailed explanation of statistics.
Results: I have minor comments.
For section 3.1 Patient characteristics. : Can authors add patient's ECOG and CCI (Charlson Co-morbidity) or any equivalent data of their performance status or comorbidity burden? This will give readers a better understanding of the study population.
In section 3.2.: I recommend adding the total number of AEs in the text. It seems some patients had more than one AEs, and to also add how many patients had more than one AEs, esp the number of patients having more than one irAEs.
Discussion: Is well written. However, I would recommend adding a paragraph on the limitation of the study.
Author Response
This is a very well-written study. This phase I study aimed to assess the safety of a WT1 peptide vaccine in combination with nivolumab in patients with WT1-expressing ovarian cancer in second or third remissions.
Introduction: Very well written. No major comments.
Methodology: Well explained. With a detailed explanation of statistics.
Response: We appreciate these comments.
Results: I have minor comments.
For section 3.1 Patient characteristics. : Can authors add patient's ECOG and CCI (Charlson Co-morbidity) or any equivalent data of their performance status or comorbidity burden? This will give readers a better understanding of the study population.
Response: Agree. We had included the Karnofsky Performance Status but have now also included the ECOG to line 271. All patients had a KPS of 90% which is the equivalent to an ECOG 0.
In section 3.2.: I recommend adding the total number of AEs in the text. It seems some patients had more than one AEs, and to also add how many patients had more than one AEs, esp the number of patients having more than one irAEs.
Response: We agree. Section 3.2 has been expanded to include more information regarding the AEs. 64% of patients had more than 1 AE. All adverse-events were considered irAEs due to the nature of the study intervention
Discussion: Is well written. However, I would recommend adding a paragraph on the limitation of the study.
Response: We agree and starting Line 422 we recognize the limitations of this study given the small study size. We acknowledge that outcomes including progression free survival were difficult to correlate with immune response data given this small sample size.
Reviewer 3 Report
Beryl L. Manning-Geist, et al., reported that phase I study results of a multivalent WT1 peptide vaccine along with nivolumab in patients with WT1-expressing ovarian cancer in second or third remission.
Major comments:
1. Authors have not explained any rationale for selecting the “nivolumab”
2. nivolumab causes non-targeted myocarditis/myositis, in that case, what is the further monitoring of that patient?
3. On Page, 4 and line 158, the authors mentioned “CA-125 levels were obtained at baseline, weeks 6, 15, and every .... for up to 1 year….” but there is no graph showing CA125 levels at various time points as they mentioned.
Minor comments:
1. It is better to expand the words like AE and ICI
Author Response
Major comments:
- Authors have not explained any rationale for selecting the “nivolumab”
Response: We agree and have now included further information at line 92. The rationale for combination therapy with immune checkpoint inhibition is to create a favorable immunologic environment by attenuating immunosuppressive mechanisms, thereby encouraging the action of effector T cells generated by these vaccines. Nivolumab has been studied in many solid tumors including ovarian cancer. It has been studied as a companion immune checkpoint inhibitor in several clinical trials due to its tolerability and potential effectiveness to enhance the efficacy and duration of response to cancer vaccines.
2. nivolumab causes non-targeted myocarditis/myositis, in that case, what is the further monitoring of that patient?
Response: We have provided additional information regarding the further monitoring of this patient at Line 307. The patient required a prolonged steroid taper and long-term rehabilitation with physical therapy. The patient thereafter remained under the care of a cardiologist for ongoing management of her permanent pacemaker. Six months after her discontinuation from the study, the patient returned to her baseline status and subsequently resumed chemotherapy.
3. OnPage, 4 and line 158, the authors mentioned “CA-125 levels were obtained at baseline, weeks 6, 15, and every .... for up to 1 year….” but there is no graph showing CA125 levels at various time points as they mentioned.
Response: We agree and this is now included as a supplementary Figure 9.
Minor comments:
- It is better to expand the words like AE and ICI
Response: We agree and these words are expanded throughout the manuscript.